# Epidemiology and Outcomes of Hypernatraemia in Patients with COVID-19—A Territory-Wide Study in Hong Kong

**DOI:** 10.3390/jcm12031042

**Published:** 2023-01-29

**Authors:** Benjamin Y. F. So, Chun Ka Wong, Gordon Chun Kau Chan, Jack Kit Chung Ng, Grace Chung Yan Lui, Cheuk Chun Szeto, Ivan Fan Ngai Hung, Hung Fat Tse, Sydney C. W. Tang, Tak Mao Chan, Kai Ming Chow, Desmond Y. H. Yap

**Affiliations:** 1Division of Nephrology, Department of Medicine, Queen Mary Hospital, The University of Hong Kong, Hong Kong SAR, China; 2Division of Cardiology, Department of Medicine, Queen Mary Hospital, The University of Hong Kong, Hong Kong SAR, China; 3Division of Nephrology, Department of Medicine and Therapeutics, Prince of Wales Hospital, The Chinese University of Hong Kong, Hong Kong SAR, China; 4Division of Infectious Diseases, Department of Medicine and Therapeutics, Prince of Wales Hospital, The Chinese University of Hong Kong, Hong Kong SAR, China; 5Division of Infectious Diseases, Department of Medicine, Queen Mary Hospital, The University of Hong Kong, Hong Kong SAR, China

**Keywords:** hypernatraemia, sodium, COVID-19, epidemiology, outcomes

## Abstract

Background: Dysnatraemias are commonly reported in COVID-19. However, the clinical epidemiology of hypernatraemia and its impact on clinical outcomes in relation to different variants of SARS-CoV-2, especially the prevailing Omicron variant, remain unclear. Methods: This was a territory-wide retrospective study to investigate the clinical epidemiology and outcomes of COVID-19 patients with hypernatraemia at presentation during the period from 1 January 2020 to 31 March 2022. The primary outcome was 30-day mortality. Key secondary outcomes included rates of hospitalization and ICU admission, and costs of hospitalization. Results: In this study, 53,415 adult COVID-19 patients were included for analysis. Hypernatraemia was observed in 2688 (5.0%) patients at presentation, of which most cases (99.2%) occurred during the local “5th wave” dominated by the Omicron BA.2 variant. Risk factors for hypernatraemia at presentation included age, institutionalization, congestive heart failure, dementia, higher SARS-CoV-2 Ct value, white cell count, C-reactive protein and lower eGFR and albumin levels (*p* < 0.001 for all). Patients with hypernatraemia showed significantly higher 30-day mortality (32.0% vs. 5.7%, *p* < 0.001) and longer lengths of stay (12.9 ± 10.9 vs. 11.5 ± 12.1 days, *p* < 0.001) compared with those with normonatraemia. Multivariate analysis revealed hypernatraemia at presentation as an independent predictor for 30-day mortality (aHR 1.32, 95% CI 1.14–1.53, *p* < 0.001) and prolonged hospital stays (OR 1.55, 95% CI 1.17–2.05, *p* = 0.002). Conclusions: Hypernatraemia is common among COVID-19 patients, especially among institutionalized older adults with cognitive impairment and other comorbidities during large-scale outbreaks during the Omicron era. Hypernatraemia is associated with unfavourable outcomes and increased healthcare utilization.

## 1. Introduction

Disorders of sodium and water balance are common in hospitalized patients, particularly the elderly [1,2]. Although hypernatraemia occurs less frequently than hyponatraemia [3,4], it is associated with dramatically increased morbidity and mortality across a wide range of medical and surgical conditions [5]. Hypernatraemia most commonly arises as a result of hypotonic fluid loss, insufficient intake of free water, or, less commonly, excess sodium intake or intoxication [6]. Under physiological conditions, the human body possesses robust regulatory mechanisms that defend against fluctuations in sodium balance via control of renal sodium and water excretion, stimulation of thirst by crosstalk with the hypothalamic–pituitary system and expression of homeostatic receptors in the skin. These mechanisms are sometimes overwhelmed in acutely ill patients, resulting in varying degrees of hypernatraemia [6]. Such derangements are particularly exaggerated in frail older adults, especially those with cognitive impairment, who are unable to compensate for ongoing fluid losses [7].

Dysnatraemias are commonly reported in COVID-19 [8]. Most reports thus far have focused on hyponatraemia, which occurs commonly among patients with COVID-19 and may be a marker of disease severity [9,10,11]. However, hypernatraemia (commonly defined as a plasma or serum sodium level of greater than 145 mmol/L0) has also been observed in COVID-19, and may be more specific than hyponatraemia for predicting poor disease outcomes in COVID-19, as shown by a recent meta-analysis including seven studies [12]. The pathophysiology of hyponatraemia and hypernatraemia in COVID-19 appears to be disparate and therefore ought to be studied independently.

Most previous reports on dysnatraemias in COVID-19, including those on hypernatraemia, were published in the pre-Omicron era [8,13,14]. However, each variant of SARS-CoV-2 may be associated with a distinct constellation of clinical symptoms and end-organ complications [15]. Furthermore, the rapidly evolving Omicron outbreak has crippled healthcare systems around the world, including in Hong Kong, leading to a sea change in the clinical phenotype of patients presenting to healthcare services with COVID-19. In Hong Kong, the “5th wave” of COVID-19 driven by the Omicron BA.2 subvariant overwhelmed the public healthcare system rapidly, with a significant proportion of the population infected, including a large number of frail nursing home residents, many of whom presented with severe, life-threatening hypernatraemia [16,17]. Here, we report on the territory-wide prevalence and clinical correlates of patients diagnosed with COVID-19 and hypernatraemia at presentation, with particular emphasis on ongoing outbreaks due to Omicron subvariants.

## 2. Materials and Methods

### 2.1. Study Design and Patient Selection

This study was a territory-wide retrospective observational cohort study. Adult patients who tested positive for SARS-CoV-2 by RT-PCR (reverse transcription polymerase chain reaction) in respiratory samples, and with serum sodium (Na) levels available on the same day from 1 January 2020 to 31 March 2022, were identified from the Clinical Data Analysis and Reporting System (CDARS) database of the Hong Kong Hospital Authority. CDARS is an electronic database that captures comprehensive clinical data of all patients registered in public hospitals and clinics in Hong Kong. Previous data validation for use in cohort studies showed high coding accuracy [18,19]. Retrieved data included patients’ demographics, institutionalization (defined by patients who utilized the service of the Community Geriatric Assessment Team, which delivers outreach service to elderly homes and institutions), diagnoses, hospitalization, prescriptions, laboratory results and deaths. All data retrieved were deidentified to ensure patient privacy and confidentiality. The disease diagnosis was cross-checked with the diagnosis coding in CDARS using the International Classification of Diseases, Ninth Revision, Clinical Modification (ICD-9-CM) (Appendix A). The estimated glomerular filtration rate (eGFR) was calculated using the CKD Epidemiology Collaboration (CKD-EPI) 2009 creatinine equation. Hypernatraemia and normonatraemia were defined as serum Na being above 145 mmol/L, and from 135 to 145 mmol/L, respectively. In Hong Kong, all patients with COVID-19 who required hospital admission were admitted to public hospitals. Treatment, including the use of antiviral and/or immunomodulatory therapies (Table 1), of patients with COVID-19 was at clinicians’ discretion and according to prevailing protocols at the time. Concurrent comorbidity load was further weighed using Charlson Comorbidity Index (CCI) [20] (Appendix A).

The study protocol was approved by the Institutional Review Board of the University of Hong Kong/Hospital Authority Hong Kong West Cluster (HKU/HA IRB UW 13-625), and the study was conducted in compliance with the Declaration of Helsinki.

### 2.2. Outcomes

All subjects were followed for at least 90 days or until death. The primary outcome was 30-day mortality following diagnosis of COVID-19. The secondary outcomes included rate of hospitalization and intensive care unit (ICU) hospitalization. In addition, we evaluated the impact of hypernatraemia on hospitalization and length of stay (LOS) among the surviving cohort. We also compared the rates of hypernatraemia among local waves driven by different SARS-CoV-2 variants (Table 2). The costs of hospitalization were estimated from the nominal daily costs of general medical and ICU beds (653.8 USD/day and 3128.2 USD/day, respectively) multiplied by the LOS in the respective beds.

### 2.3. Statistical Analysis

Statistical analysis was performed using SPSS for Mac software version 27.0 (IBM corporation, Armonk, NY, USA). Continuous data were expressed as mean ± standard deviation, while categorical data were presented as number (percentage). Patients were grouped according to the presence/absence of hypernatraemia at presentation for analysis. Data were compared between groups using chi-square test, Student’s *t*-test or Mann–Whitney U test as appropriate. Time-to-event analysis was performed for the primary outcome using the Kaplan–Meier method and compared using the log-rank test. Furthermore, multivariate logistic and Cox proportional hazard regression analysis were performed to adjust for confounders. Factors known to affect COVID-19 outcomes and clinical parameters significantly different between patients with hyper- and normonatraemia were adjusted for in the multivariate analysis model. A *p*-value below 0.05 was considered statistically significant. All probabilities were two-tailed.

## 3. Results

### 3.1. Patient Characteristics

The data from a total of 53,415 adult patients were retrieved and included for final analysis (Figure 1). A total of 2688 (5.0%) adult patients with COVID-19 had hypernatraemia on presentation, while 36,182 (67.7%) had normonatraemia. A total of 14,545 (27.2%) patients who had hyponatraemia at presentation were excluded from the comparative analysis to avoid skewing the results. The clinical characteristics of COVID-19 patients with hypernatraemia or normonatraemia at presentation, and their hospitalization, ICU admission and treatment data are summarized in Table 3 and Table 4. 

Among the hypernatraemic patients, a baseline sodium level within 6 months of the index hospitalization was available for 2118 (78.8%). The mean prehospitalization sodium level was 139.6 ± 3.4 mmol/L. Only 76 (3.6%) patients had pre-existing hypernatraemia. Patients with hypernatraemia were older (86.3 ± 10.5 years vs. 62.4 ± 22.0, *p* < 0.001) and more likely to be institutionalized (67.8% vs. 18.1%, *p* < 0.001). Before adjustment for baseline variables, these patients had higher SARS-CoV-2 viral load (Ct values 23.0 ± 6.4 vs. 23.5 ± 6.8, *p* = 0.006), C-reactive protein (11.0 ± 8.7 mg/L vs. 3.7 ± 5.9 mg/L, *p* < 0.001), creatine kinase (405 ± 1527 U/L vs. 250 ± 1599 U/L, *p* < 0.001) and D-dimer levels (1886 ± 2594 ng/mL vs. 862 ± 1567 ng/mL, *p* < 0.001) on univariate analysis (Table 3). They were more likely to receive immunomodulatory therapy (58.7% vs. 23.0%, *p* < 0.001) during the disease course, though the antiviral agent utilization was lower (21.7% vs. 25.3%, *p* < 0.001) (Table 4). COVID-19 patients with hypernatraemia at presentation had higher CCI than those with normonatraemia (2.71 ± 2.20 vs. 1.41 ± 1.92, *p* < 0.001). Among components of CCI, dementia (39.9% vs. 9.3%, *p* < 0.001), diabetes mellitus (29.5% vs. 19.9%, *p* < 0.001) and cerebrovascular accident (20.2% vs. 7.9%, *p* < 0.001) were more frequent in patients with hypernatraemia (Table 5). They also presented with higher white cell and neutrophil counts, but lower lymphocyte counts and haemoglobin levels (*p* < 0.001 for all). eGFR (43.6 ± 26.5 mL/min vs. 79.7 ± 31.1 mL/min/1.73 m^2^, *p* < 0.001) and serum albumin levels (29.3 ± 6.2 g/L vs. 36.8 ± 6.4 g/L, *p* < 0.001) were lower in COVID-19 patients with hypernatraemia compared with those with normonatraemia.

Most COVID-19 cases with hypernatraemia (99.2%) occurred during the “5th wave”, driven by the Omicron BA.2 variant (Table 6). The incidence rate of hypernatraemia was significantly higher during the “5th wave” compared with previous local waves (6.2% vs. 0.2%, *p* < 0.001) (Table 6 and Table 7).

### 3.2. Predictors of Hypernatraemia in COVID-19 Patients

Multivariate analysis showed that age, institutionalization, congestive heart failure, dementia, higher SARS-CoV-2 Ct value (thus, lower viral loads), lower haemoglobin, higher white cell count, higher C-reactive protein and lower eGFR and lower albumin levels were associated with a higher risk of hypernatraemia in COVID-19 infection, after adjusting for confounding factors (Table 4).

### 3.3. Mortality

A total of 4390 of the 53,415 patients had died at 30 days of follow-up (pooled mortality rate of 8.2%). The 30-day mortality rate was significantly higher in the hypernatraemic group compared with normonatraemic controls (32.0% vs. 5.7%, *p* < 0.001) (Table 6 and Figure 2). Patients who died had a higher incidence rate of hypernatraemia at presentation (19.6% vs. 3.7%, *p* < 0.001), accompanied by higher mean plasma Na levels at presentation (138.7 ± 10.1 vs. 136.8 ± 6.1 mmol/L, *p* < 0.001) (Table 8). Patients who died were older, had more comorbidities (CCI, 2.82 ± 2.32 vs. 1.58 ± 1.99, *p* < 0.001) and showed a higher prevalence of institutionalization (45.1% vs. 20.6%, *p* < 0.001) (Table 8). The rates of antiviral (26.9% vs. 27.8%, *p* = 0.02) and immunomodulatory (26.2% vs. 67.2%, *p* < 0.001) therapy use were lower in patients who eventually died. Multivariate analysis demonstrated hypernatraemia at presentation as an independent predictor for 30-day mortality (adjusted hazard ratio (aHR) 1.32, 95% CI 1.14–1.53, *p* < 0.001) (Table 9).

### 3.4. Impact on Healthcare Utilization

We analysed healthcare utilization in surviving patients with hypernatraemia or normonatraemia at presentation. There was no difference in the hospitalization rates between patients with hypernatraemia and normonatraemia (62.9% vs. 64.0%, *p* = 0.2). However, the overall LOS was longer (12.9 ± 10.9 vs. 11.5 ± 12.1 days, *p* < 0.001) among surviving patients with hypernatraemia, with a greater proportion of patients with prolonged hospitalization (i.e., >14 days) (35.2% vs. 26.2%, *p* < 0.001) (Table 6 and Table 10). Multivariate analysis revealed hypernatraemia at presentation as an independent predictor for prolonged hospitalization (i.e., LOS > 7 days) in COVID-19 (odds ratio (OR) 1.55, 95% CI 1.17–2.05, *p* = 0.002). Other predictors identified from the same model include institutionalization (OR 1.27, 95% CI 1.06–1.52, *p* = 0.009), SARS-CoV-2 PCR Ct value (OR 0.94, 95% CI 0.93–0.94, *p* < 0.001), the presence of chronic liver disease (OR 1.45, 95% CI 1.13–1.86, *p* = 0.004), biochemical parameters such as white cell count (OR 0.97, 95% CI 0.95–0.98, *p* < 0.001), eGFR (OR 0.99, 95% CI 0.99–0.99, *p* = 0.001), albumin (OR 1.02, 95% CI 1.01–1.03, *p* = 0.002), C-reactive protein (OR 1.04, 95% CI 1.03–1.06, *p* < 0.001) and the need for COVID-19 treatment including antiviral (OR 1.44, 95% CI 1.27–1.64, *p* < 0.001) and immunomodulatory therapies (OR 1.57, 95% CI 1.35–1.82, *p* < 0.001) (Table 10). Among patients with hypernatraemia who survived to hospital discharge, those who required intensive care unit care had a 5.5-fold higher overall cost of hospitalization than those managed solely in general wards (USD 18,141 (IQR 4730-31,552) vs. USD 5558 (IQR 2289-8827), *p* < 0.001). Nonetheless, the cost of hospitalization did not differ between patients with mild, moderate and severe hypernatraemia at presentation.

## 4. Discussion

In this territory-wide retrospective cohort study involving 53,415 patients with COVID-19, we observed a substantial rate of hypernatraemia at presentation to hospital, especially during the “5th wave” caused by the Omicron BA.2 subvariant in Hong Kong. COVID-19 patients with hypernatraemia at presentation generally showed worse clinical outcomes, with significantly increased 30-day mortality. Patients with hypernatraemia at presentation who survived their acute hospital stay tended to have longer LOS, and accrued higher healthcare costs. Importantly, COVID-19 patients with hypernatraemia at presentation were overwhelmingly elderly, and a significant proportion of them were institutionalized, in stark contrast to those with normonatraemia.

The rate of hypernatraemia in COVID-19 appears to be context-specific, and can be significantly affected by patient characteristics, healthcare settings and infection control policies. During the earliest waves of COVID-19 in the spring of 2020, the prevalence of hypernatraemia in Hong Kong was merely 0.1% (Table 7). During the same period, in which the outbreak was all driven by the same ancestral strain of COVID, hypernatraemia was reported in 3.7% and 9.1% of COVID-19 patients in Europe and the United States, respectively [8,13]. The meticulous case tracking and mass quarantine practiced in Hong Kong at the time enabled early detection of cases with mild to moderate symptoms and hospitalization of virtually all positive cases. The prevalence of hypernatraemia surged to 6.2% when the healthcare system was overwhelmed by the “5th wave” (caused by the Omicron BA.2 subvariant) in Hong Kong [16,17,23]. COVID-19 patients, especially the elderly, often presented late to medical care after a protracted waiting time at home or in nursing homes, during which they developed dehydration and hypernatraemia. The finding that advanced age, institutionalization and dementia were predictors for hypernatraemia in COVID-19 patients lends further support to our postulation. After adjustment for demographic variables and other risk factors, an inverse relationship between viral load and hypernatraemia was observed, suggesting that these patients might be late presenters, when viral shedding was already waning. Physical and neurocognitive inability to compensate for ongoing insensible fluid losses in these elderly institutionalized patients likely contributed to the development of hypernatraemia.

Our results highlight that hypernatraemia during large COVID-19 outbreaks is a symptom of an overburdened, dysfunctional healthcare system. Hypernatraemia and its associated adverse outcomes can potentially be prevented or mitigated if at-risk individuals are closely monitored and given adequate fluid replacement. This is particularly important as we identified hypernatraemia as a strong predictor of mortality in our cohort, even after adjusting for other comorbidities. In a large European registry, hypernatraemia predicted mortality and development of sepsis [8]. A registry analysis from New York showed that inpatient mortality was particularly increased in patients with severe hypernatraemia complicating COVID-19 [13]. Hypernatraemia per se does not appear to be pathogenic in COVID-19; in fact, some experimental studies suggest that therapeutic induction of hypernatraemia may protect against lung injury [24,25,26,27,28]. Instead, we speculate that hypernatraemia during acute illnesses may be a surrogate marker of frailty, especially in the geriatric population. The close correlation between hypernatraemia in COVID-19 and excess mortality was likely exaggerated in this group of patients with a background of frailty, compounded with poor oral fluid and food intake during acute illness. The role of medications such as diuretics remains to be further elucidated.

There are several limitations in this territory-wide observational cohort study. First, owing to the retrospective observational nature of this study, a definitive causal relationship between hypernatraemia and mortality could not be determined. Whether mortality related to hypernatraemia could be mitigated by appropriate fluid management remains speculative, as only the sodium level on initial presentation was captured in the analysis, and serial values were not fully analysed. Second, due to the constraints of this registry analysis, certain clinical variables, including vital signs, disease severity scores or frailty indices were not available for most patients. Although hypernatraemia is classically associated with dehydration, a significant proportion of hypernatraemic patients could in fact be hypervolaemic, especially in the critically ill population [29]; however, fluid status could not be determined with confidence in our cohort. Third, reporting bias may occur as the registry analysis mostly captures patients who were hospitalized or who reported their diagnosis to the official reporting system. Fourth, hypernatraemia may be masked by other biochemical abnormalities, especially hyperglycaemia [30]. As paired plasma glucose and sodium levels were not available for all patients, there is a possibility that the rate of hypernatraemia may have been underestimated.

These limitations notwithstanding, this study’s key strength lies in its large sample size, with over 50,000 patients with COVID-19 analysed with a specific focus on hypernatraemia. All patients were followed for at least 90 days or until death, allowing for evaluation of various key short- to medium-term outcomes. Second, since all patients in our study were diagnosed by RT-PCR performed on upper respiratory tract specimens, we were able to examine the correlations between the viral loads and clinical outcomes to determine if there was a genuine causal link between infection per se and development of hypernatraemia. Finally, with data available from different waves of COVID-19 in Hong Kong, we were able to delineate longitudinal trends in the prevalence of hypernatraemia among presenting patients. Based on these trends, we surmise that the rate of hypernatraemia can be highly variable during different outbreaks of COVID-19, depending both on the demographics of the populations affected and the robustness of the healthcare system.

## 5. Conclusions

Hypernatraemia at presentation is associated with excess mortality and prolonged hospitalization among COVID-19 patients. Advanced age, dementia and institutionalization are important risk factors for hypernatraemia in COVID-19 patients. An inverse relationship between viral load of SARS-CoV-2 and hypernatraemia suggests that these patients often present late to healthcare services, highlighting a key area for improvement.

## Figures and Tables

**Figure 1 jcm-12-01042-f001:**
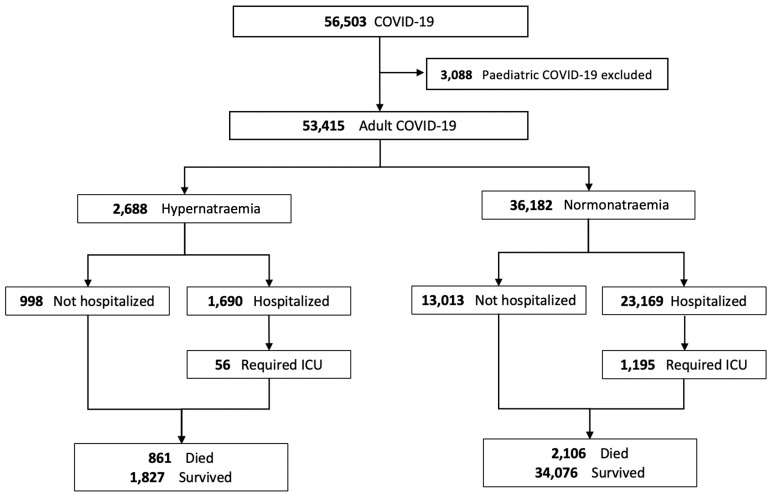
Disposition of patients with COVID-19 and the relationship with blood sodium levels.

**Figure 2 jcm-12-01042-f002:**
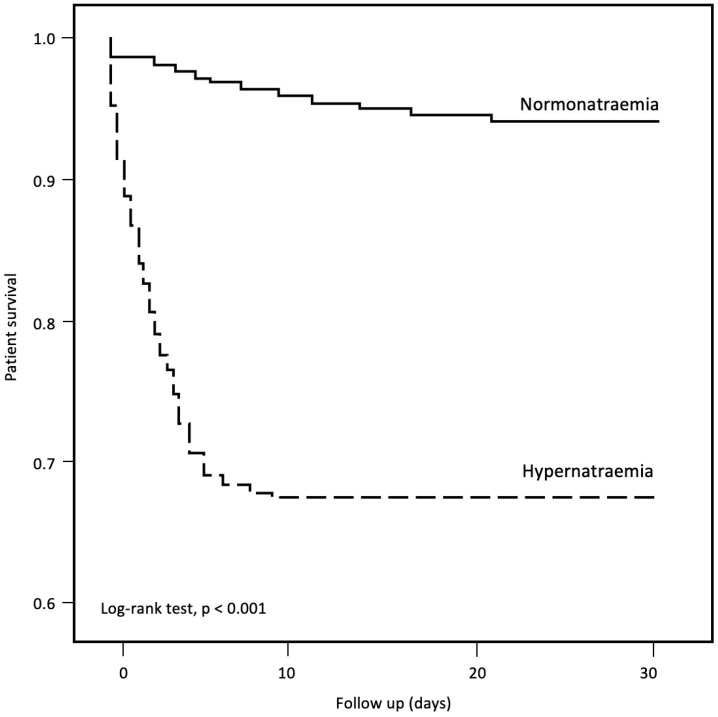
Thirty-day mortality in COVID-19 patients with hypernatraemia and normonatraemia.

**Table 1 jcm-12-01042-t001:** Antiviral and immunomodulatory therapies used in Hong Kong for COVID-19.

Antiviral Therapy	Immunomodulatory Therapy
Interferon beta-1b Lopinavir/Ritonavir Molnupiravir Nirmatrelvir/Ritonavir Ribavirin Remdesivir	Baricitinib Dexamethasone Tocilizumab

**Table 2 jcm-12-01042-t002:** Time period and dominant SARS-CoV-2 variant during each local breakthrough wave during COVID-19.

Wave	Time Period	Dominant SARS-CoV-2 Variant
2nd	1–30 April 2020	D614G [21]
3rd	15 June–30 September 2020	B.1.1.63 [22]
4th	1 November 2022–28 February 2021	B.1.36.27 [22]
5th	1 January–31 March 2022	Omicron BA.2

**Table 3 jcm-12-01042-t003:** Clinical characteristics of COVID-19 patients with hypernatraemia or normonatraemia at presentation.

	Hypernatraemia (*n* = 2688)	Normonatraemia (*n* = 36,182)	*p*-Value
Age	86.3 ± 10.5	62.4 ± 22.0	<0.001 ^a^
Age older than 65, No. (%)	2550 (94.9%)	17,679 (48.9%)	<0.001 ^b^
Male, No. (%)	1300 (48.4%)	18,103 (50.0%)	0.095 ^b^
Institutionalized, No. (%)	1823 (67.8%)	6533 (18.1%)	<0.001 ^b^
Charlson Comorbidity Index	2.71 ± 2.20	1.41 ± 1.92	<0.001 ^a^
Major comorbidities			
Diabetes mellitus	793 (29.5%)	7184 (19.9%)	<0.001 ^b^
Hypertension	1691 (62.9%)	13,618 (37.6%)	<0.001 ^b^
Ischaemic heart disease	452 (16.8%)	3724 (10.3%)	<0.001 ^b^
Cerebrovascular accident	544 (20.2%)	2847 (7.9%)	<0.001 ^b^
Cardiac arrhythmia	497 (18.5%)	3858 (10.7%)	<0.001 ^b^
Congestive heart failure	367 (13.7%)	2676 (7.4%)	<0.001 ^b^
Chronic obstructive airway disease	159 (5.9%)	1604 (4.4%)	<0.001 ^b^
Asthma	50 (1.9%)	593 (1.6%)	0.4 ^b^
Pneumoconiosis	38 (1.4%)	242 (0.7%)	<0.001 ^b^
Dementia	1072 (39.9%)	3380 (9.3%)	<0.001 ^b^
Chronic liver disease	208 (7.7%)	2026 (5.6%)	<0.001 ^b^
Active malignancy	576 (17.7%)	5517 (15.2%)	0.001 ^b^
Chronic kidney disease			<0.001 ^b^
Stage 1	67 (2.5%)	12,037 (33.3%)	
Stage 2	842 (31.3%)	16,915 (46.7%)	
Stage 3	715 (26.6%)	4860 (13.4%)	
Stage 4	616 (22.9%)	1435 (4.0%)	
Stage 5	448 (16.7%)	934 (2.6%)	
Laboratory parameters			
SARS-CoV-2 RT-PCR Ct value on admission	23.0 ± 6.4	23.5 ± 6.8	0.006 ^a^
Haemoglobin (g/dL)	11.8 ± 2.6	12.8 ± 2.2	<0.001 ^a^
White cell count (10^9^/L)	11.3 ± 7.4	7.0 ± 4.0	<0.001 ^a^
Neutrophil (10^9^/L)	7.3 ± 4.5	4.9 ± 3.4	<0.001 ^a^
Lymphocyte (10^9^/L)	1.0 ± 1.4	1.3 ± 1.0	<0.001 ^a^
Neutrophil to lymphocyte ratio	13.5 ± 13.1	5.6 ± 7.2	<0.001 ^a^
Platelet (10^9^/L)	231 ± 104	223 ± 88	0.05 ^a^
Sodium (mmol/L)	153.2 ± 7.0	138.6 ± 2.3	<0.001 ^a^
Potassium (mmol/L)	4.1 ± 0.9	3.9 ± 0.5	<0.001 ^a^
Urea (mmol/L)	21.8 ± 13.6	6.7 ± 6.1	<0.001 ^a^
Creatinine (µmol/L)	188 ± 167	103 ± 124	<0.001 ^a^
eGFR (by CKD-EPI equation) (mL/min/1.73 m^2^)	43.6 ± 26.5	79.7 ± 31.1	<0.001 ^a^
Albumin (g/L)	29.3 ± 6.2	36.8 ± 6.4	<0.001 ^a^
C-reactive protein (mg/L)	11.0 ± 8.7	3.7 ± 5.9	<0.001 ^a^
Calcium (mmol/L)	2.25 ± 0.23	2.23 ± 0.15	<0.001 ^a^
Phosphate (mmol/L)	1.26 ± 0.57	1.09 ± 0.37	<0.001 ^a^
Plasma osmolality (mOsm/kg)	354 ± 29	302 ± 33	<0.001 ^a^
Thyroid-stimulating hormone (mIU/L)	1.3 ± 2.9	1.7 ± 3.8	0.049 ^a^
Creatine kinase (U/L)	405 ± 1527	250 ± 1599	<0.001 ^a^
D-dimer (ng/mL)	1886 ± 2594	862 ± 1567	<0.001 ^a^
Urine sodium (mmol/L)	47.7 ± 32.2	50.3 ± 40.7	0.7 ^a^
Urine osmolality (mOsm/kg)	559 ± 148	438 ± 196	<0.001 ^a^

Data are presented as mean ± standard deviation unless specified and compared using Student’s *t*-test ^a^ and chi-square test ^b^. COVID-19, novel coronavirus disease-2019; SARS-CoV-2, severe acute respiratory syndrome coronavirus 2; RT-PCR, reverse transcription polymerase chain reaction; Ct value, cycle threshold value; eGFR, estimated glomerular filtration rate; CKD-EPI, Chronic Kidney Disease-Epidemiology Collaboration.

**Table 4 jcm-12-01042-t004:** Predictors for hypernatraemia at presentation in patients with COVID-19.

	Univariate Model		Multivariate Model	
	OR (95% CI)	*p*-Value	OR (95% CI)	*p*-Value
Demographics				
Age	1.09 (1.08–1.09)	<0.001	1.03 (1.03–1.04)	<0.001
Institutionalization	9.57 (8.78–10.42)	<0.001	2.37 (2.00–2.82)	<0.001
SARS-CoV-2 RT-PCR Ct value	0.99 (0.99–0.99)	0.006	1.04 (1.02–1.05)	<0.001
Comorbidities				
CHF	1.98 (1.76–2.23)	<0.001	0.76 (0.59–0.97)	0.03
Dementia	6.44 (5.91–7.01)	<0.001	1.80 (1.50–2.14)	<0.001
Laboratory parameters				
Haemoglobin	0.84 (0.83–0.86)	<0.001	1.15 (1.11–1.20)	<0.001
White cell count	1.17 (1.16–1.18)	<0.001	1.06 (1.04–1.07)	<0.001
eGFR (by CKD-EPI equation)	0.96 (0.96–0.96)	<0.001	0.97 (0.97–0.97)	<0.001
C-reactive protein	1.11 (1.11–1.12)	<0.001	1.02 (1.01–1.03)	<0.001
Albumin	0.86 (0.85–0.86)	<0.001	0.92 (0.91–0.94)	<0.001

CHF, congestive heart failure; CI, confidence interval; CKD-EPI, Chronic Kidney Disease-Epidemiology Collaboration; COVID-19, novel coronavirus disease-2019; Ct value, cycle threshold value; eGFR, estimated glomerular filtration rate; RT-PCR, reverse transcription polymerase chain reaction; SARS-CoV-2, severe acute respiratory syndrome coronavirus 2.

**Table 5 jcm-12-01042-t005:** Charlson Comorbidity Index and its components in COVID-19 patients with hypernatraemia or normonatraemia at presentation.

	Hypernatraemia (*n* = 2688)	Normonatraemia (*n* = 36,182)	*p*-Value
Charlson Comorbidity Index Score	2.71 ± 2.20	1.41 ± 1.92	<0.001 ^a^
Components of Charlson Comorbidity Index			
Acute myocardial infarction	452 (16.8%)	3724 (10.3%)	<0.001 ^b^
Congestive heart failure	367 (13.7%)	2676 (7.4%)	<0.001 ^b^
Peripheral vascular disease	16 (0.6%)	102 (0.3%)	0.004 ^b^
Cerebrovascular disease	544 (20.2%)	2847 (7.9%)	<0.001 ^b^
Dementia	1072 (39.9%)	3380 (9.3%)	<0.001 ^b^
Chronic lung disease	159 (5.9%)	1604 (4.4%)	<0.001 ^b^
Rheumatic disease	362 (13.5%)	3743 (10.3%)	<0.001 ^b^
Peptic ulcer	253 (9.4%)	1662 (4.6%)	<0.001 ^b^
Mild liver disease	191 (7.1%)	1833 (5.1%)	<0.001 ^b^
Moderate to serious liver disease	19 (0.7%)	217 (0.6%)	0.5 ^b^
Mild to moderate diabetes	793 (29.5%)	7184 (19.9%)	<0.001 ^b^
Diabetes with chronic complications	341 (12.7%)	2338 (6.5%)	<0.001 ^b^
Hemiplegia or paraplegia	156 (5.8%)	803 (2.2%)	<0.001 ^b^
Kidney disease	460 (17.1%)	1041 (2.9%)	<0.001 ^b^
Malignancy	451 (16.8%)	5090 (14.1%)	<0.001 ^b^
Solid, metastatic tumour	24 (0.9%)	433 (1.2%)	0.2 ^b^
Leukaemia	5 (0.2%)	54 (0.1%)	0.6 ^b^
Lymphoma	7 (0.3%)	104 (0.3%)	0.8 ^b^
AIDS	4 (0.1%)	23 (0.1%)	0.1 ^b^

Data are presented as mean ± standard deviation unless specified and compared using Student’s *t*-test ^a^ and chi-square test ^b^.

**Table 6 jcm-12-01042-t006:** Clinical outcomes in COVID-19 patients with hypernatraemia or normonatraemia at presentation and relationship with different local waves.

	Hypernatraemia (*n* = 2688)	Normonatraemia (*n* = 36,182)	*p*-Value
Death within 30 days	860 (32.0%)	2051 (5.7%)	<0.001 ^b^
Local wave (Time periods; dominant SARS-CoV-2 variant)			<0.001 ^b^
2nd wave (1 to 30 April 2020; D614G [21])	1 (0.1%)	746 (92.7%)	
3rd wave (15 June–30 September 2020; B.1.1.63 [22])	12 (0.4%)	2808 (90.1%)	
4th wave (1st November 2020–28 February 2021; B.1.36.27 [22])	7 (0.1%)	4514 (88.9%)	
5th wave (1 January–31 March 2022; Omicron BA.2)	2667 (6.2%)	26,484 (66.2%)	
COVID-19 Treatments			
Antiviral therapy	584 (21.7%)	9168 (25.3%)	<0.001 ^b^
Immunomodulatory therapy	1577 (58.7%)	8333 (23.0%)	<0.001 ^b^
Healthcare utilization in surviving patients	Hypernatraemia (n = 1827)	Normonatraemia (n = 34,076)	*p*-Value
Duration of hospitalization	12.9 ± 10.9	11.5 ± 12.1	<0.001 ^a^
Hospitalization for > 14 days	334 (35.2%)	5540 (26.2%)	<0.001 ^b^
ICU admission	40 (4.2%)	1043 (4.9%)	0.3 ^b^
Duration of ICU admission	7.9 ± 19.5	7.8 ± 13.0	0.9 ^a^
ICU hospitalization for > 7 days (%, among hospitalized in ICU)	9 (22.5%)	294 (28.2%)	0.4 ^b^

Data are presented as mean ± standard deviation unless specified and compared using Student’s *t*-test ^a^ and chi-square test ^b^. COVID-19, coronavirus disease-2019; ICU, intensive care unit; SARS-CoV-2, severe acute respiratory syndrome coronavirus 2.

**Table 7 jcm-12-01042-t007:** Clinical characteristics of COVID-19 patients with hypernatraemia at presentation during the different local waves of COVID-19.

	2nd Wave (*n* = 1)	3rd Wave (*n* = 12)	4th Wave (*n* = 7)	5th Wave (*n* = 2667)	*p*-Value
Age	75	68.3 ± 15.8	82.4 ± 15.2	86.4 ± 10.3	<0.001 ^a^
Age older than 65, No. (%)	1 (100%)	7 (58.3%)	6 (85.7%)	2536 (95.1%)	<0.001 ^b^
Male, No. (%)	0 (0%)	5 (41.7%)	3 (42.9%)	1292 (48.4%)	0.7 ^b^
Institutionalized, No. (%)	0 (0%)	4 (33.3%)	3 (42.9%)	1816 (68.1%)	0.01 ^b^
SARS-CoV-2 RT-PCR Ct value	34.7	25.1 ± 7.9	25.6 ± 6.2	23.1 ± 6.4	0.1 ^a^
Charlson Comorbidity Index	0	2.33 ± 2.77	3.14 ± 2.41	2.71 ± 2.19	0.5 ^a^
Comorbidities, No. (%)					
Diabetes mellitus	0 (0%)	3 (25.0%)	2 (28.6%)	788 (29.6%)	0.9 ^b^
Hypertension	0 (0%)	6 (50.0%)	4 (57.1%)	1681 (63.0%)	0.4 ^b^
Ischaemic heart disease	0 (0%)	1 (8.3%)	4 (57.1%)	447 (16.8%)	0.06 ^b^
Cerebrovascular accident	0 (0%)	2 (16.7%)	0 (0%)	542 (20.3%)	0.7 ^b^
Cardiac arrhythmia	0 (0%)	1 (8.3%)	2 (28.6%)	494 (18.5%)	0.8 ^b^
Congestive heart failure	0 (0%)	0 (0%)	2 (28.6%)	365 (13.7%)	0.5 ^b^
COAD	0 (0%)	0 (0%)	0 (0%)	159 (6.0%)	0.9 ^b^
Asthma	0 (0%)	0 (0%)	0 (0%)	50 (1.9%)	1.0 ^b^
Pneumoconiosis	0 (0%)	0 (0%)	0 (0%)	38 (1.4%)	1.0 ^b^
Dementia	0 (0%)	2 (16.7%)	1 (14.3%)	1069 (40.1%)	0.2 ^b^
Chronic liver disease	0 (0%)	1 (8.3%)	2 (28.6%)	129 (4.8%)	0.07 ^b^
Active malignancy	0 (0%)	3 (25.0%)	2 (28.6%)	470 (17.6%)	0.8 ^b^
Chronic kidney disease, No. (%)					<0.001 ^b^
Stage 1	0 (0%)	6 (50.0%)	0 (0%)	60 (2.2%)	
Stage 2	0 (0%)	3 (25.0%)	2 (28.6%)	837 (31.4%)	
Stage 3	1 (100%)	1 (8.3%)	2 (28.6%)	711 (26.7%)	
Stage 4	0 (0%)	1 (8.3%)	3 (42.9%)	612 (22.9%)	
Stage 5	0 (0%)	1 (8.3%)	0 (0%)	447 (16.8%)	
Laboratory parameters					
Haemoglobin (g/dL)	10.4	12.9 ± 2.4	11.5 ± 2.2	11.8 ± 2.6	0.6 ^a^
White cell count (10^9^/L)	18.7	6.8 ± 1.8	9.6 ± 5.1	11.3 ± 7.5	0.2 ^a^
Neutrophil (10^9^/L)	15.7	4.3 ± 1.5	6.5 ± 3.2	9.5 ± 5.7	0.006 ^a^
Lymphocyte (10^9^/L)	0.7	1.8 ± 0.9	1.3 ± 1.4	1.0 ± 1.4	0.4 ^a^
Neutrophil to lymphocyte ratio	23.1	3.4 ± 3.1	18.3 ± 18.9	13.6 ± 13.1	0.05 ^a^
Platelet (10^9^/L)	183	246 ± 78	186 ± 66	231 ± 104	0.8 ^a^
Potassium (mmol/L)	4.0	3.8 ± 0.6	4.2 ± 1.0	4.1 ± 0.9	0.7 ^a^
Urea (mmol/L)	22.2	8.3 ± 5.8	19.0 ± 12.2	21.9 ± 13.6	0.007 ^a^
Creatinine (umol/L)	118.0	92.7 ± 64.7	142.1 ± 54.7	188.5 ± 167.9	0.3 ^a^
eGFR (by CKD-EPI) (mL/min/1.73 m^2^)	39.0	76.9 ± 31.5	41.7 ± 24.7	43.5 ± 26.4	<0.001 ^a^
Albumin (g/L)	21.0	37.8 ± 5.8	30.5 ± 6.0	29.3 ± 6.1	<0.001 ^a^
C-reactive protein (mg/L)	8.4	2.5 ± 4.0	6.6 ± 7.8	11.1 ± 8.7	0.008 ^a^
Calcium (mmol/L)	2.26	2.28 ± 0.14	2.07 ± 0.14	2.25 ± 0.23	0.3 ^a^
Phosphate (mmol/L)	1.30	0.99 ± 0.15	1.51 ± 1.04	1.27 ± 0.57	0.5 ^a^
Plasma osmolality (mOsm/kg)	355	357 ± 26	361 ± 28	354 ± 29	0.8 ^a^
Thyroid stimulating hormone (mIU/L)	2.8	3.8 ± 7.4	1.3 ± 1.7	1.3 ± 2.8	0.09 ^a^
D-dimer (ng/mL)	253	247 ± 116	315 ± 625	1892 ± 2597	0.3 ^a^

Data are presented as mean ± standard deviation unless specified and compared using Student’s *t*-test ^a^ and chi-square test ^b^. CKD-EPI, Chronic Kidney Disease-Epidemiology Collaboration; COAD, chronic obstructive airway disease; Ct value, cycle threshold value; eGFR, estimated glomerular filtration rate; RT-PCR, reverse transcription polymerase chain reaction; SARS-CoV-2, severe acute respiratory syndrome coronavirus 2.

**Table 8 jcm-12-01042-t008:** Clinical characteristics of COVID-19 patients who died within 30 days.

	Died (*n* = 4318)	Survived (*n* = 49,025)	*p*-Value
Age	83.2 ± 11.5	65.4 ± 21.3	<0.001 ^a^
Age older than 65, No. (%)	3979 (92.1%)	27,137 (55.3%)	<0.001 ^b^
Male, No. (%)	2596 (60.1%)	25,044 (51.0%)	<0.001 ^b^
Institutionalized, No. (%)	1972 (45.7%)	10,099 (20.6%)	<0.001 ^b^
Serum sodium (mmol/L)	138.7 ± 10.2	136.8 ± 6.1	<0.001 ^a^
Hypernatraemia, No. (%)	860 (29.5%)	1828 (5.1%)	<0.001 ^b^
Charlson Comorbidity Index	2.8 ± 2.3	1.6 ± 2.0	<0.001 ^a^
Comorbidities			
Diabetes mellitus	1470 (34.0%)	11,475 (23.4%)	<0.001 ^b^
Hypertension	2707 (62.7%)	20,512 (41.8%)	<0.001 ^b^
Ischaemic heart disease	859 (19.9%)	5562 (11.3%)	<0.001 ^b^
Cerebrovascular accident	822 (19.0%)	4395 (9.0%)	<0.001 ^b^
Cardiac arrhythmia	986 (22.8%)	5610 (11.4%)	<0.001 ^b^
Congestive heart failure	760 (17.6%)	3755 (7.6%)	<0.001 ^b^
Chronic obstructive airway disease	414 (9.6%)	2303 (4.7%)	<0.001 ^b^
Asthma	72 (1.7%)	850 (1.7%)	0.8 ^b^
Pneumoconiosis	111 (2.6%)	370 (0.8%)	0.001 ^b^
Dementia	1093 (25.3%)	5091 (10.4%)	<0.001 ^b^
Chronic liver disease	376 (8.7%)	2891 (5.9%)	<0.001 ^b^
Active malignancy	905 (21.0%)	8090 (16.5%)	<0.001 ^b^
Chronic kidney disease			<0.001^b^
Stage 1	244 (5.7%)	14,401 (29.3%)	
Stage 2	1513 (35.0%)	23,548 (48.0%)	
Stage 3	1192 (27.6%)	7252 (14.8%)	
Stage 4	762 (17.6%)	2205 (4.5%)	
Stage 5	607 (14.1%)	1691 (3.4%)	
COVID-19 treatments			
Antiviral therapy	1200 (27.8%)	13,220 (26.9%)	0.02 ^b^
Immunomodulatory therapy	2919 (67.6%)	12,375 (26.2%)	<0.001 ^b^

Data are presented as mean ± standard deviation unless specified and compared using Student’s *t*-test ^a^ and chi-square test ^b^. COVID-19, coronavirus disease-2019.

**Table 9 jcm-12-01042-t009:** Risk factors for 30-day mortality in patients with COVID-19.

	Univariate Model		Multivariate Model	
	HR (95% CI)	*p*-Value	Adjusted HR (95% CI)	*p*-Value
Hypernatraemia	6.97 (6.44–7.55)	<0.001	1.32 (1.14–1.53)	<0.001
Demographics				
Age	1.06 (1.06–1.06)	<0.001	1.03 (1.02–1.04)	<0.001
Male sex	1.35 (1.27–1.43)	<0.001	1.18 (1.04–1.34)	0.01
Comorbidities				
Charlson Comorbidity Index	1.26 (1.24–1.27)	<0.001		
Diabetes mellitus	1.71 (1.61–1.83)	<0.001		
Hypertension	2.37 (2.22–2.52)	<0.001		
Ischaemic heart disease	1.97 (1.83–2.12)	<0.001		
Cerebrovascular accident	2.31 (2.15–2.50)	<0.001		
COAD	2.07 (1.87–2.29)	<0.001	1.50 (1.22–1.83)	<0.001
Active malignancy	1.56 (1.45–1.68)	<0.001		
Dementia	2.62 (2.45–2.81)	<0.001		
Congestive heart failure	2.48 (2.29–2.68)	<0.001		
Arrhythmia	2.28 (2.13–2.45)	<0.001	1.22 (1.05–1.42)	0.01
Chronic liver disease	1.50 (1.35–1.67)	<0.001		
Laboratory parameters				
SARS-CoV-2 RT-PCR Ct value	0.97 (0.96–0.97)	<0.001	0.98 (0.97–0.99)	<0.001
Haemoglobin	0.79 (0.78–0.80)	<0.001	0.96 (0.93–0.99)	0.004
White cell count	1.02 (1.02–1.02)	<0.001	1.01 (1.00–1.02)	0.006
eGFR (by CKD-EPI)	0.97 (0.97–0.97)	<0.001	0.99 (0.99–0.99)	<0.001
Albumin	0.88 (0.87–0.88)	<0.001	0.96 (0.95–0.97)	<0.001
C-reactive protein	1.10 (1.10–1.11)	<0.001	1.05 (1.05–1.06)	<0.001
D-dimer (every 1000 units rise)	1.22 (1.20–1.23)	<0.001	1.03 (1.00–1.06)	0.04
COVID-19 Treatment				
Antiviral therapy	0.75 (0.70–0.80)	<0.001		
Immunomodulatory therapy	4.43 (4.15–4.72)	<0.001	2.20 (1.88–2.58)	<0.001

CI, confidence interval; CKD-EPI, Chronic Kidney Disease-Epidemiology Collaboration; COAD, chronic obstructive airway disease; COVID-19, coronavirus disease-2019; Ct value, cycle threshold value; eGFR, estimated glomerular filtration rate; RT-PCR, reverse transcription polymerase chain reaction; SARS-CoV-2, severe acute respiratory syndrome coronavirus 2.

**Table 10 jcm-12-01042-t010:** Risk factors for prolonged hospitalization (i.e., >7 days) among surviving patients with COVID-19.

	Univariate Model		Multivariate Model	
	OR (95% CI)	*p*-Value	Adjusted OR (95% CI)	*p*-Value
Hypernatraemia	1.44 (1.24–1.66)	<0.001	1.55 (1.17–2.05)	0.002
Demographics				
Age	0.99 (0.99–1.00)	0.007		
Male sex	1.17 (1.11–1.23)	<0.001		
Institutionalization	1.16 (1.09–1.24)	<0.001	1.27 (1.06–1.52)	0.009
SARS-CoV-2 RT-PCR Ct value	0.93 (0.93–0.93)	<0.001	0.94 (0.93–0.94)	<0.001
Comorbidities				
Dementia	1.15 (1.06–1.26)	0.001		
Chronic liver disease	1.02 (1.92–1.14)	0.7	1.45 (1.13–1.86)	0.004
Laboratory parameters				
Haemoglobin	1.06 (1.05–1.07)	<0.001		
White cell count	0.97 (0.97–0.98)	<0.001	0.97 (0.95–0.98)	<0.001
eGFR (by CKD-EPI)	0.99 (0.99–1.00)	0.02	0.99 (0.99–0.99)	0.001
Albumin	1.01 (1.00–1.01)	0.002	1.02 (1.01–1.03)	0.002
C-reactive protein	1.02 (1.02–1.03)	<0.001	1.04 (1.03–1.06)	<0.001
Treatment for COVID-19				
Antiviral therapy	1.98 (1.88–2.09)	<0.001	1.44 (1.27–1.64)	<0.001
Immunomodulatory therapy	1.93 (1.82–2.05)	<0.001	1.57 (1.35–1.82)	<0.001

CI, confidence interval; CKD-EPI, Chronic Kidney Disease-Epidemiology Collaboration; COVID-19, coronavirus disease-2019; Ct value, cycle threshold value; eGFR, estimated glomerular filtration rate; RT-PCR, reverse transcription polymerase chain reaction; SARS-CoV-2, severe acute respiratory syndrome coronavirus 2.

## Data Availability

The data presented in this study are available on request from the corresponding author. The data are not publicly available due to institution-level internal policies.

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
