# Peer review of "Epidemiology and Outcomes of Hypernatraemia in Patients with COVID-19—A Territory-Wide Study in Hong Kong"

_jcm, 2023, doi:10.3390/jcm12031042_

Round 1

Reviewer 1 Report

The authors focused this study on serum sodium levels in patients with COVID-19. The paper itself is well-structured and worth submitting.

According to the present cohort study, 67.7% had normal Na levels, 5.0% had high Na levels, and 27.2% had low Na levels.
If this study were conducted to elucidate the association between serum Na level and the prognosis of COVID-19, 27.2% of patients with low Na levels should not be excluded from this study.

And it is necessary to verify whether serum Na was normal or already abnormal before the onset of COVID-19.

I would like to see a statement in the text regarding the above two points.

Author Response

Thank you for the kind and helpful comments.

Regarding the 27.2% of patients who were hyponatraemic, in our preliminary analysis we realized that the risk factors and demographic profile of patients with hyponatraemia was vastly different from those with hypernatraemia. Since hyponatraemia had already been covered extensively in the literature, we decided to focus this particular article on hypernatraemia. Thus, hyponatraemic patients were excluded from the comparative analysis to avoid skewing the results (we have added a comment to this effect in the text as well).

We agree that it is necessary to differentiate between patients who had pre-existing hypernatraemia and those who developed it de novo after COVID-19 infection. In the updated text (line 165) we explain and illustrate that the vast majority of hypernatraemic cases only developed hypernatramia after COVID-19 infection. 

Reviewer 2 Report

The authors present their retrospective analysis of 53,415 adult patients with COVID-19 infection, who had a serum Na obtained at admission.   The presence of hypernatremia (Na above 145 mmol/L) in 5% of patients conferred a higher 30-day mortality rate and a longer length of stay, when compared to patients with normonatremia.    

The authors performed an analysis of a large data set of biochemical values. I suggest defining hypernatremia earlier, for instance in the Introduction. Suggest specifying the version year of the CKD-EPI formula, 2012 or 2021, since in 2021 there was a new formula adopted by the nephrology community. The cycle threshold data (Ct values) are a bit confusing – 23.0±6.4 for hypernatremia group and 23.5±6.8 for normonatremia group. They comment in the Results section line 163, that the patients with hypernatremia had a higher viral load – this needs to be revised. With respect to the mean Na during hospitalization and rate of mortality, I would suggest caution with drawing conclusions, since the serum Na depends on the fluid management during that period. Useful would be a mention of weight changes, if any, and presence of edema. Even in normonatremic patients, etiology of CKD is important, for example: salt-losing nephropathies, diuretic use in CHF, etc, and consider the fact that hyperglycemia may mask hypernatremia. In the Discussion section, the authors state as one the strengths (line 324) the determination of causality between infection and sequalae of hypernatremia – however, this is not clear from the study, as there may be sequalae due to COVID-19 infection.   

Author Response

Thank you for the kind comments and suggestions.

As suggested, we will now include a commonly used definition of hypernatraemmia in the introduction sectlon.

The 2009 version of the CKD-EPI equation was used as data collection started in 2020, before the promulgation of the latest equation.

The viral load of hypernatraemic patients was slightly higher (with a lower Ct value) on univariate analysis, but the effect was abrogated on multivariate analysis and in fact they had lower viral load (higher Ct value) after adjustment.

We agree with caution in overinterpreting the link between hypernatraemia and mortality. We address this specifically in our discussion section. The need to see serial measurements of sodium, rather than looking at a single screenshot of a point in time, is also explained. Changes in weight and assessment fluid status (including the presence of edema) was unfortunately impossible in this registry analysis and we also explain this in the discussion section. We note, and cite in the article, that a significant proportion of hypernatraemic patients in critically ill populations may in fact be hypervolaemic. The effects of other confounders, such as concomitant hyperglycaemia, as well the role of drugs including diuretics, is touched upon in the updated text; unfortunately, we were unable to draw definitive conclusions in these aspects.

The previous sentence on causality in the discussion was poorly worded. We have now updated the text to reflect the fact that although the lack of a definite link between viral load and hypernatraemia argues against direct causality by COVID-19 infection, it still cannot be entirely ruled out. This study is hypothesis-generating and we suggest that other aspects including underlying frailty, institutionalization and multiple comorbidities are the real causes of hypernatraemia.